# Chemical modulation of gut bacterial metabolism induces colanic acid and extends the lifespan of nematode and mammalian hosts

Guo Hu[ORCID][1], Marzia Savini[2], Matthew Brandon Cooke[1], Xin Wei[3], Dinghuan Deng[2], Shihong M. Gao[2], Ruyue Alps Xia[4], Youchen Guan[2], Alice X. Wen[1], Xin Yu[5], Jin Wang[5], Chao Jiang[3], Christophe Herman[1], Jiefu Li[2], Meng C. Wang[ORCID][2]*

1 Department of Molecular and Human Genetics, Baylor College of Medicine, Houston, Texas, United States of America, 2 Janelia Research Campus, Howard Hughes Medical Institute, Ashburn, Virginia, United States of America, 3 Life Sciences Institute, Zhejiang University, Hangzhou, Zhejiang, China, 4 Riverside High School, Ashburn, Virginia, United States of America, 5 Department of Pharmacology and Chemical Biology, Baylor College of Medicine, Houston, Texas, United States of America

* mengwang@janelia.hhmi.org

## Abstract

Microbiota-derived metabolites have emerged as key regulators of longevity. The metabolic activity of the gut microbiota, influenced by dietary components and ingested chemical compounds, profoundly impacts host fitness. While the benefits of dietary prebiotics are well-known, chemically targeting the gut microbiota to enhance host fitness remains largely unexplored. Here, we report a novel chemical approach to induce a pro-longevity bacterial metabolite in the host gut. We discovered that wild-type *Escherichia coli* strains overproduce colanic acids (CAs) when exposed to a low dose of cephaloridine, leading to an increased life span in the host organism *Caenorhabditis elegans*. In the mouse gut, oral administration of low-dose cephaloridine induced transcription of the capsular polysaccharide synthesis (*cps*) operon responsible for CA biosynthesis in commensal *E. coli* at 37 °C, and attenuated age-related metabolic changes. We also found that low-dose cephaloridine overcomes the temperature-dependent inhibition of CA biosynthesis and promotes its induction through a mechanism mediated by the membrane-bound histidine kinase ZraS, independently of cephaloridine's known antibiotic properties. Our work lays a foundation for microbiota-based therapeutics through chemical modulation of bacterial metabolism and highlights the promising potential of leveraging bacteria-targeting drugs in promoting host longevity.

## Introduction

The gut microbiota present in the gastrointestinal (GI) tract plays a crucial role in human health and disease susceptibility [1], influencing the host's neuronal functions [2], immunity [3], and life expectancy [4]. The microbiota genetic composition (termed

**Data availability statement:** Bacterial transcriptome sequencing data is available through the Sequence Read Archive under the accession number PRJNA1111054. Metagenomic sequencing data is available under the accession number PRJNA1265280. All relevant data generated or analyzed in this study are included in this manuscript and/or its supplementary information as source data. Code Availability The code utilized for high-resolution bacteria imaging is available on GitHub (https://github.com/huguo0519/Cepha-project.git), also see in Supporting information, S3 Data. The code utilized for analyzing the cps operon is included in Supporting information, S4 Data.

**Funding:** The research of this work was supported by Howard Hughes Medical Institute (M.C.W.). C.H. was supported by NIH DP1AI152073. The funders had no role in study design, data collection and analysis, decision to publish, or prepration of the manuscript.

**Competing interests:** I have read the journal's policy and the authors of this manuscript have the following competing interests: A patent application has been filed related to the work described in this manuscript, with M.C.W. and G.H. listed as inventors. J.W. is the co-founder of Chemical Biology Probes LLC. J. W. has stock ownership in CoRegen Inc and serves as a consultant for this company. J.W. and X.Y. are the co-founders of Fortitude Biomedicines Inc. and hold equity interest in this company.

**Abbreviations:** ANOVA, analysis of variance; CA, colanic acid; Cepha-CAI dose, colanic acid-inducing dose; cps, capsular polysaccharide synthesis; EtOH, ethanol; GI, gastrointestinal; HDL, high-density lipoprotein; IACUC, Institutional Animal Care and Use Committee; LB, Luria–Broth; LDL, low-density lipoprotein; MSA, multiple sequence alignment; NEFA, non-esterified free fatty acid; NGM, nematode growth medium; PBP, penicillin-binding protein; SRS, stimulated Raman scattering; sub-MIC, sub-minimum inhibitory concentrations, TCS, two-component system.

microbiome) of over 2000 genera [5] encodes crucial enzymes that mediate the production of unique microbial metabolites, host-derived secondary metabolites, and host-microbe-shared metabolites [6]. These microbial metabolites provide energy to local intestinal epithelial cells as essential nutrients [7] and influence remote organs through blood circulation as signaling molecules [8]. The microbiota also responds to environmental inputs, such as changes in diet and medication uses, to produce bioactive compounds based on ingested diets [9] or alter drug efficacy to be more active, less active, or even toxic [10], resulting in both advantageous and disadvantageous outcomes for the host. The interaction between drugs and gut microbiota plays a crucial role in a variety of biological processes. One such process is the xenobiotic effect, where the microbiota converts prodrugs into active or inactive forms, thereby altering their bioavailability and bioactivity [11]. Conversely, exposure to pharmaceuticals can influence metabolic pathways within the microbiota by modifying microbial community composition and regulating enzyme activities. Studies in both mice and humans have demonstrated that GI exposure to various drugs leads to changes in the composition of the gut microbiota [12–14]. Commonly used medications, such as proton pump inhibitors, metformin, antibiotics, and laxatives, have been linked to alterations in microbial metabolic pathways [15]. However, the precise mechanisms underlying these drug-induced changes, as well as their functional consequences on gut physiology and host health, remain poorly understood.

In recent years, an increasing number of microbiome-based therapeutics have shown benefits to the host, including fecal microbiota transplantation, dietary prebiotics, enteral reconstitution of symbiotic bacteria, the introduction of engineered bacteria, and supplementation of microbiota-derived bioactive compounds [16]. Here, we introduce a microbiome-based approach that uses host-impermeable drugs to specifically target metabolic biosynthesis pathways in gut commensals to promote host longevity. Distinct from those existing strategies, this approach chemically targets the existing commensal community to induce the production of metabolic products beneficial to the host. We demonstrated this approach in both *Caenorhabditis elegans* (*C. elegans*) and *Mus musculus* (mice), and further uncovered the molecular mechanism underlying this chemical-induced effect in *Escherichia coli* (*E. coli*).

## Results

### Low-dose cephaloridine induces CA beyond temperature restriction to prolong life span

Colanic acid (CA) is an extracellular polysaccharide synthesized in *E. coli* and other *Enterobacteriaceae* [17]. Its biosynthesis is catalyzed by a series of enzymes encoded in the capsular polysaccharide synthesis (*cps*) operon [18]. Our previous studies have revealed the longevity-promoting effect of CA in *C. elegans* and *Drosophila melanogaster* [19,20]. We also showed that the optogenetic induction of the *cps* operon is sufficient to prolong *C. elegans* life span [20]. To examine the relevance of the *cps* operon in the human gut microbiome, we have searched the Unified Human Gastrointestinal Genome (UHGG) [21] using DIAMOND [22]. In this database, 47 out of 4,744 species (approximately 0.99%) contain at least one genome

that carries the *cps* operon, corresponding to 5,578 genomes out of 289,232 total genomes. These include 4,598 in *Escherichia*, 466 in *Salmonella*, 341 in *Enterobacter*, 147 in *Citrobacter*, and 26 in other species (Fig 1A).

Next, we analyzed two publicly available metagenomic datasets of human centenarian gut microbiome (PRJNA675598, PRJEB25514) [23,24]. A recent study reported that members of the *Enterobacteriaceae* family typically comprise less than 1% of the healthy human gut microbiota [25]. In our analysis, *Enterobacteriaceae* accounted for approximately 1.43% of total reads in the Italian cohort (PRJEB25514) and 3.02% in the Japanese cohort (PRJNA675598). We further compared the presence of the *cps* operon between centenarians and their elderly controls in these two cohorts. On average, 9.16% of total reads in the Japanese cohort and 7.25% in the Italian cohort mapped to the representative *cps*-positive genomes from 47 identified species.

In the Japanese cohort, which includes over 100 individuals per group, we observed a statistically significant difference in *cps* operon abundance between centenarians and controls (Fig 1B, Welch's *t* test, $p = 0.0071$), with a small-to-moderate effect size (Cohen's $d = 0.37$). In the Italian cohort, which includes approximately 20 individuals per group, a statistically significant difference was also observed (Fig 1C, Welch's *t* test, $p = 0.0436$), with a moderate effect size (Cohen's $d = 0.64$). Notably, a post hoc power analysis based on this effect size suggests that approximately 39 individuals per group would be required to achieve 80% power at $\alpha = 0.05$, indicating that the Italian dataset may have been underpowered in the original analysis. Despite this limitation, these findings collectively support a small-to-modest increase of *cps* operon abundance in centenarian individuals. Unfortunately, neither cohort studies included information on individual diet, medication use, or lifestyle factors, preventing us from controlling these potential confounding variables. Moreover, this association analysis cannot clarity whether the increased *cps* operon abundance contributes to longevity or is simply a consequence of advanced age in centenarians. Nevertheless, these observations motivated us to investigate whether the *cps* operon can be induced in human gut-derived commensal bacteria.

*Escherichia* represents the largest genus carrying the *cps* operon, and *E. coli* is known to be among the first commensal bacteria to inhabit the human gut [26]. However, CA production from the *E. coli cps* operon becomes restricted when the environmental temperature exceeds 30 °C [27]. A previous study indicated that certain β-lactam antibiotics trigger the overexpression of the *cps* operon in *E. coli* at sub-minimum inhibitory concentrations (sub-MIC) at 37 °C [28]. We thus utilized an *E. coli* strain that harbors a LacZ reporter for the *cps* operon transcriptional regulation to screen multiple β-lactam antibiotics and antibiotics that disrupt the outer membrane, protein synthesis, and genome replication. Using a colorimetric assay, we examined the levels of β-galactosidase from the transcriptional up-regulation of the *cps* operon, resulting in intensified blue coloration. We confirmed that cephaloridine (Cepha) and cefazolin (Cafez) strongly induce the *cps* operon, while cephalothin (CephT), cefuroxime (Cefu), and penicillins—benzylpenicillin (PenG), ampicillin (Amp), and carbenicillin (Carb) show various levels of moderate effects based on the blue coloration (Figs 1D and S1A).

Next, we quantified CA production in the bacteria culture medium upon the treatment of different antibiotics. The medium from *E. coli* treated with cephaloridine at sub-MIC showed the highest CA level, while penicillin and cephalothin showed lower or no increase (Fig 1E). Furthermore, we found that the most effective CA-inducing concentration of cephaloridine was 1.8 μg/mL (Fig 1F). Unlike the growth delay commonly observed with sub-MIC antibiotic exposure [29], wild-type *E. coli* treated with 1.8 μg/mL cephaloridine exhibited growth rates comparable to untreated controls and to CA-overproducing Δ*lon* mutants [19] (Fig 1G).

We referred to this low dose of cephaloridine as the Colanic Acid-Inducing dose (Cepha-CAI dose).

We further examined whether the treatment at the Cepha-CAI dose affects *E. coli* metabolic activity. To this end, we have applied stimulated Raman scattering (SRS) microscopy to monitor the incorporation of deuterium-labeled glucose into individual bacteria, enabling assessment of their metabolic activity and viability at single-cell resolution. Compared to the control, the Cepha-CAI treatment did not alter glucose incorporation during the 1–6 h growth window (Figs 1H, 1I, and S1B). Together, these results demonstrate that the Cepha-CAI dose overrides the temperature restriction of CA production at 37 °C without inhibiting *E. coli* growth or causing cell inviability.

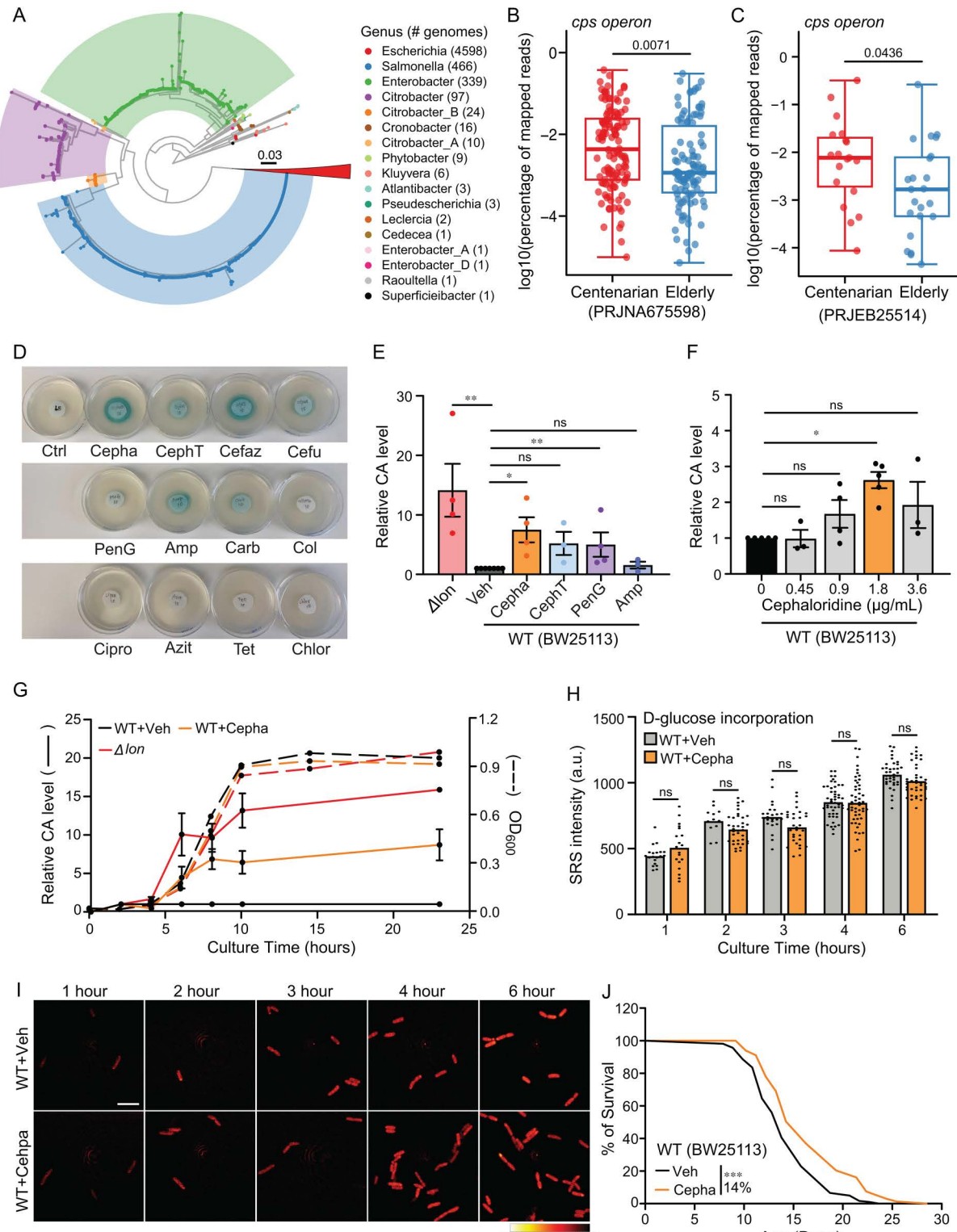

**Fig 1. Chemical induction of bacterial colanic acid (CA) promotes longevity. (A)** Phylogenetic tree of the *cps* operon from 5,578 genomes across 47 species within the gut microbiota. Tree leaves are colored by genus, with the legend on the right showing the number of genomes for each genus. The largest clade, comprising 4,577 *E. coli* genomes, is collapsed into a red triangle. In addition to *Escherichia*, dominant clades from *Salmonella*,

*Enterobacter*, and *Citrobacter* are highlighted in their respective genus-specific colors. Also see the tree file in S2 Data. **(B)** Comparison of the percentage of reads mapping to the *cps* operon between centenarians and elderly individuals in the Japanese cohort (PRJNA675598). The *p* value was obtained using the Welch's *t* test. **(C)** Same as (B), but showing the comparison in the Italian cohort (PRJEB25514). **(D)** The disk-diffusion assay shows *E. coli* treated by chemical compounds presenting various blue colorations, indicating the induction of the *cps* operon responsible for CA biosynthesis. Ctrl (LB medium), Cepha (cephaloridine), CephT (cephalothin), Cefaz (Cefazolin), Cefu (cefuroxime), PenG (benzylpenicillin), Amp (ampicillin), Carb (carbenicillin), Col (colistin), Cipro (ciprofloxacin), Azit (Azithromycin), Tet (tetracycline), and Chlor (chloramphenicol); *N*=2. **(E)** Quantification of CA from filtered bacterial cultures shows the lon deletion mutant *E. coli* (Δlon) produces an increased level of CA by 14-fold as compared to the wild-type (WT) BW25113 *E. coli*. Cepha at 1.8 μg/mL and PenG at 5 μg/mL induce CA production by 7-fold and 5-fold, respectively; CephT at 0.625 μg/mL and Amp at 1.5 μg/mL do not induce CA significantly ($p > 0.05$). *N*≥3 **(F)** Quantification of the relative CA induction in WT BW25113 *E. coli* when responding to 0, 0.45, 0.9, 1.8, and 3.6 μg/mL Cepha. A low dose of cephaloridine at 1.8 μg/mL is defined as the Colanic Acid-Inducing dose (Cepha-CAI dose), *N*≥3. **(G)** Bacterial growth curves for WT *E. coli* with the vehicle and Cepha-CAI dose treatments and the Δlon mutant. WT *E. coli* treated by Cepha-CAI dose and the Δlon mutant *E. coli* increase CA levels during the log phase (solid lines) and do not affect bacterial growth (dotted lines). *N*=2 biologically independent replicates. **(H)** Measurement of carbon-deuterium levels in WT *E. coli* cultured in M9 medium supplemented with deuterium-labeled glucose using stimulated Raman scattering (SRS) microscopy at 2180 cm$^{-1}$, indicating no metabolic suppression by the Cepha-CAI dose. Independent biological replicates are shown in S1 Fig. **(I)** Representative SRS images of WT *E. coli* cultured with deuterium-labeled glucose, scale bar=1 μm. **(J)** The life span of *C. elegans* is extended by 14% with Cepha-CAI dose-treated WT BW25113 *E. coli*. *N*=3, 60–100 worms per replicate. In (E) and (F), error bars represent the mean±standard error of the mean (s.e.m.). **$p < 0.01$, *$p < 0.05$, ns $p > 0.05$ by two-tailed Student *t* test. In (J), ***$p < 0.001$ by log-rank test, also seen in S1 Table. In (H), Error bars represent the mean±standard deviation (s.d.), ns $p > 0.05$ by Two-way ANOVA. (B, C, E, F, G, H, J) Source data available in S1 Data.

We further revealed that, similar to the Δ*lon* mutant, wild-type *E. coli* treated by Cepha-CAI dose showed increased CA production during the log phase, and the elevated levels persisted into the stationary phase (Fig 1G). Importantly, wild-type *C. elegans* with *E. coli* treated with the Cepha-CAI dose showed a 14% extension in life span compared to the controls without the treatment (Fig 1J and S1 Table). This demonstrates that chemically induced CA effectively confers longevity-promoting benefits in the host *C. elegans*.

## CA-induction by low-dose cephaloridine improves metabolic health during mouse aging

Due to its poor oral absorption, cephaloridine as an antibiotic agent was terminated in the clinic [30]. Even at a concentration of 1 mM, cephaloridine cannot be sufficiently absorbed by the GI tract due to a high efflux rate [30]. Since we successfully induced CA using the Cepha-CAI dose that bypassed the temperature constraint at 37 °C, we hypothesized that cephaloridine would specifically target commensal *E. coli* and stimulate CA production in the mammalian gut, where the average body temperature ranges from 36 °C to 40 °C. To test this hypothesis, we first treated MG1655, a K-12 *E. coli* strain derived from human microbiota, with the Cepha-CAI dose at 37 °C and detected increased CA levels in the culture medium (Fig 2A). Furthermore, wild-type *C. elegans* hosting the Cepha-CAI dose-treated MG1655 showed a 13% life span extension (Fig 2B and S1 Table).

To monitor the induction of CA biosynthesis within the mouse gut, we genetically engineered the MG1655 strain to express GFP under the control of the *cps* promoter and RFP under a constitutive promoter *syn135* [31] (*cps*G-R reporter, Fig 2C). In this strain, constitutively expressed RFP indicates the baseline transcription, while inducible GFP reflects the transcriptional level of the *cps* operon. We found that the Cepha-CAI dose increases the GFP-to-RFP intensity ratio from 0.87 to 1.05 in the *cps*G-R reporter strain at 37 °C (Fig 2D and 2E). Next, we introduced this *cps*G-R reporter strain to wild-type C57B6 (B6) mice that were raised in a specific pathogen-free environment through drinking water to examine chemical induction in the murine gut. After supplementing *cps*G-R reporter *E. coli* through drinking water for 3 days, we administered cephaloridine or other drugs to mice in a single low dose through oral gavage, followed by supplementing drug-containing water for 3 h (Fig 2F), accounting and allowing for sufficient intestinal transit time [32] and *E. coli* turnover time [33].

Leveraging the minimal intestinal absorption and utilizing low doses of cephaloridine (2 μg/mL or 4.8 μM, 4 μg/mL or 9.6 μM, and 8 μg/mL or 19.2 μM), we examined its effects on gut bacteria in inducing the *cps* operon. By imaging the

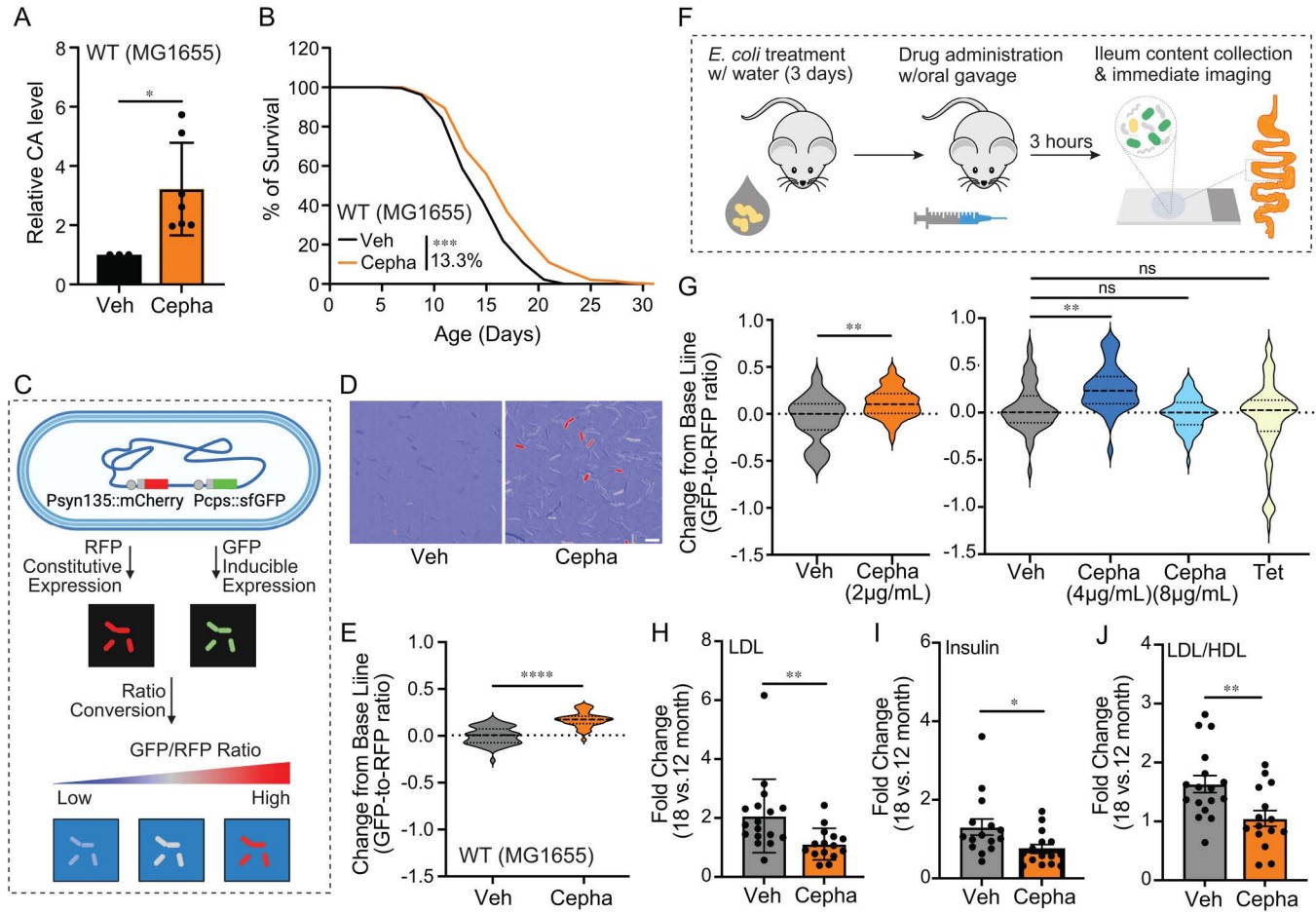

**Fig 2. A low dose of cephaloridine up-regulates colanic acid (CA) biosynthesis operon in the mouse gut microbiota. (A)** Cepha-CAI dose-treated MG1655 *E. coli* showed increased CA levels. $N > 3$. **(B)** Cepha-CAI dose-treated MG1655 *E. coli* increases *C. elegans* life span by 13.3%. $N = 3$, 60–100 worms per replicate. **(C)** Schematic diagram showing the analysis using the *cps* G-R reporter *E. coli* strain. **(D)** Representative ratio matric images of *E. coli* treated with vehicle or Cepha-CAI dose, scale bar = 5 μm. **(E)** Compared to the vehicle-treated control, Cepha-CAI dose-treated *cps* G-R reporter strain increase the GFP-to-RFP ratio from 0.87 to 1.05 (0.18 increase from the baseline). **(F)** Schematic diagram presenting the drug administration procedure in wild-type B6 mice. **(G)** Compared to the vehicle control, cephaloridine increases the GFP-to-RFP ratio of the *cps* G-R reporter *E. coli* harvested from the mouse gut from 2.01 to 2.16 at 2 μg/mL (orange) and from 2.19 to 2.41 at 4 μg/mL (dark blue), with 0.15 and 0.22 increases from the baseline, respectively. The cephaloridine at 8 μg/mL (light blue) and tetracycline treatment at 10 μg/mL (yellow) made no significant changes on the GFP-to-RFP ratio. **(H)** Plasma levels of low-density lipoprotein (LDL) increase by 2-fold between 12 and 18 months of age in control male mice, but remain nearly unchanged in mice treated orally with cephaloridine (2 μg/mL). **(I)** Between 12 and 18 months of age in female mice, the increase in plasma insulin levels is lower in mice treated orally with cephaloridine (2 μg/mL) compared to controls. **(J)** The age-related increase in the LDL/HDL ratio is suppressed by the cephaloridine treatment in male mice. In (A), error bars represent the mean ± s.e.m., * $p < 0.05$ by Student *t* test. In (B), ***$p < 0.001$ by log-rank test, also seen in S1 Table. In (E) and (G), black bars show the median of the data set, *$p < 0.05$, **$p < 0.01$, ***$p < 0.001$, ****$p < 0.0001$, ns $p > 0.05$ by Student *t* test. In (H–J), error bars represent the mean ± s.e.m., **$p < 0.01$, *$p < 0.05$, ns, $p > 0.05$ by *t* test with Welch's correction. (A, B, E, G, H–J) Source data available in S1 Data.

RFP-labeled *E. coli* harvested from the luminal content of mouse ileum, we observed increased GFP-to-RFP ratios in mice treated with 2 μg/mL (2.01 to 2.16) and 4 μg/mL (2.19 to 2.41) cephaloridine (Figs 2G and S2A). This increase was not observed in mice treated with 8 μg/mL cephaloridine (2.19 to 2.14) (Figs 2G and S2A). In the supplement group with tetracycline (10 μg/mL) that showed no CA induction (Fig 1D), we observed that the GFP-to-RFP ratio was not increased (2.19 to 2.08) (Figs 2G and S2A). These findings suggest that a low dose of cephaloridine effectively stimulates the *cps* operon of *E. coli* in the mammalian gut microbiota.

Next, we treated one-year-old wild-type mice with 2 µg/mL cephaloridine through drinking water for 6 months (S2B Fig) and profiled a series of metabolic parameters in plasma samples (S2C–S2K Fig). We found that low-density lipoprotein (LDL) levels were increased in 18-month-old male mice (Figs 2H and S2C), whereas insulin levels were elevated in 18-month-old female mice (Figs 2I and S2H). Notably, these age-related increases were attenuated by the cephaloridine treatment (Fig 2H and 2I). Furthermore, the cephaloridine treatment did not affect high-density lipoprotein (HDL) levels (S2D Fig), thereby leading to a reduction in the LDL/HDL ratio (Fig 2J). Together, these findings suggest that cephaloridine-induced CA production mitigates age-related metabolic dysfunction in mice.

We also collected fecal samples, and metagenomic analysis of these samples revealed that microbiome richness was not reduced by the cephaloridine treatment (S2L–S2N Fig), supporting the idea that low-dose cephaloridine does not cause bacterial lethality in the microbiota.

## Chemical induction of colanic acid independent of RcsC-RcsD

Next, we investigated the molecular mechanism underlying the CA induction by the Cepha-CAI dose. At nonrestricted temperatures below 30 °C, the induction of CA biosynthesis is known to be mediated by the RCS two-component system (TCS) [18]. This system involves the RcsC sensor histidine kinase situated in the inner membrane, which autophosphorylates in response to environmental cues. The RcsD phosphotransferase interacts with RcsC to transmit the phosphoryl group to the cytosolic response regulator, RcsB (Fig 3A). RcsA, interacting with RcsB, enhances the heterodimers' binding with the promoter and functions as transcription factor together to up-regulate the *cps* operon, thereby stimulating transcriptional activation (Fig 3A).

To assess the necessity of the RCS system in the chemical induction of CA at 37 °C, we treated the *E. coli* deletion mutants of *rcsA*, *rcsB*, *rcsC,* and *rcsD* with the Cepha-CAI dose. By measuring CA levels in their respective culture mediums, we found that deleting either *rcsA* or *rcsB* completely abolished the CA induction by the Cepha-CAI dose (Fig 3B). To our surprise, deleting either *rcsC* or *rcsD* did not suppress the induction (Fig 3B). These results suggest that cephaloridine relies on the RcsA-RcsB transcriptional control of the *cps* operon but operates independently of the RcsC-RcsD TCS on the bacterial inner membrane.

Further investigation of the host life span in *C. elegans* revealed that, unlike cephaloridine-treated wild-type *E. coli* (Fig 1J), the cephaloridine-treated *rcsA* deletion mutant failed to extend the host's life span (Fig 3C and S1 Table). In contrast, the cephaloridine-treated *rcsD* deletion mutant remained effective in causing life span extension (Fig 3D and S1 Table). These results further support that the chemical induction of CA in bacteria underlies the pro-longevity effect of low-dose cephaloridine on the host and suggest that the RcsC-RcsD inner membrane kinase complex is dispensable for the cephaloridine-mediated induction of pro-longevity CA.

## PBP1a and ZraS mediate the chemical induction of CA

We then hypothesized that the Cepha-CAI dose acts through other bacterial inner membrane proteins. Considering β-lactam antibiotics show preferential affinities to various penicillin-binding proteins (PBPs) [34,35], we initially explored the role of these inner membrane proteins in regulating the chemical induction of CA. In *E. coli*, cephaloridine, cefazolin, and cephalothin primarily bind to PBP1a, a protein encoded by *mrcA* [36]. This contrasts with penicillin and ampicillin, which exhibit a higher affinity for PBP4, a protein encoded by *DacB* [35] (Fig 4A). Notably, the Cepha-CAI dose failed to induce CA biosynthesis in the *mrcA* deletion mutant (ΔPBP1a), but this dose remained effective in the *dacB* deletion mutant (ΔPBP4) (Fig 4B). Thus, low-dose cephaloridine specifically acts through PBP1a to induce CA. These findings also support that the inducing effect of the Cepha-CAI dose on CA biosynthesis is distinct from the general β-lactam antibiotic effect mediated through any PBPs.

In parallel, we conducted comprehensive transcriptional profiling using wild-type MG1655 *E. coli* in response to the Cepha-CAI dose. Using RNA-seq analysis, we identified genes with significant differential expression between the

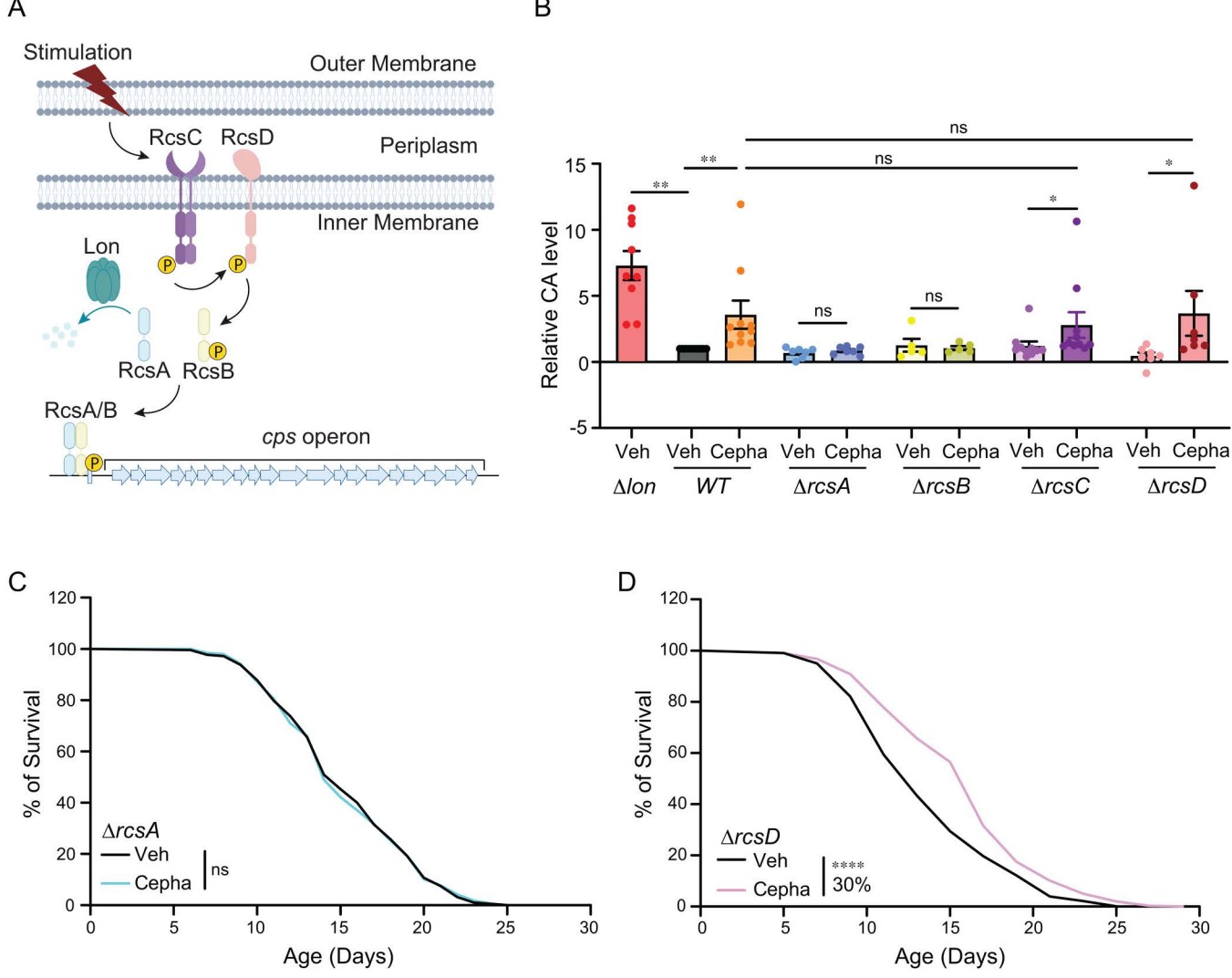

**Fig 3. Chemical induction of bacterial colanic acid (CA) acts independently of RCS inner membrane sensors. (A)** A schematic diagram of the canonical RCS activation to induce the *cps* operon and CA biosynthesis. **(B)** Treatments with Cepha-CAI dose increase CA levels in WT, ΔrcsC, and ΔrcsD mutant *E. coli* but not in the ΔrcsA or ΔrcsB mutant *E. coli*. *N* > 3. **(C)** Cepha-CAI dose-treated—ΔrcsA mutant *E. coli* fails to increase *C. elegans* life span. *N* = 3, 60–100 worms per replicate. **(D)** Cepha-CAI dose-treated—ΔrcsD mutant *E. coli* increases *C. elegans* life span by 30%. *N* = 3, 60-100 worms per replicate. In (B), error bars represent mean ± s.e.m., **$p < 0.01$, * $p < 0.05$, ns $p > 0.05$, by two-tailed Student *t* test. In (C) and (D), **** $p < 0.0001$, ns $p > 0.05$ by log-rank test, also seen in S1 Table. (B, C, D) Source data available in S1 Data.

Cepha-CAI dose-treated *E. coli* and the untreated control (FDR-adjusted *p* value < 0.05, Fig 4C). Among these genes, we observed an enrichment of RcsB target genes (Fig 4D), confirming the pivotal role of the RcsB transcription factor in the cephaloridine-mediated induction of CA. Furthermore, integrating information from transcription factor and TCS databases from RegulonDB led to the identification of four additional TCSs—BtsS-BtsR, RstB-RstA, GlrK-GlrR, and ZraS-ZraR—whose target genes exhibited enrichments among the differentially expressed candidate genes in response to the Cepha-CAI dose (Fig 4D).

To investigate the potential involvement of these TCSs, we treated the following single-gene deletion mutant *E. coli* of *btsS*, *glrK*, and *zraS* with the Cepha-CAI dose at 37 °C, and then measured the CA levels in their respective culture

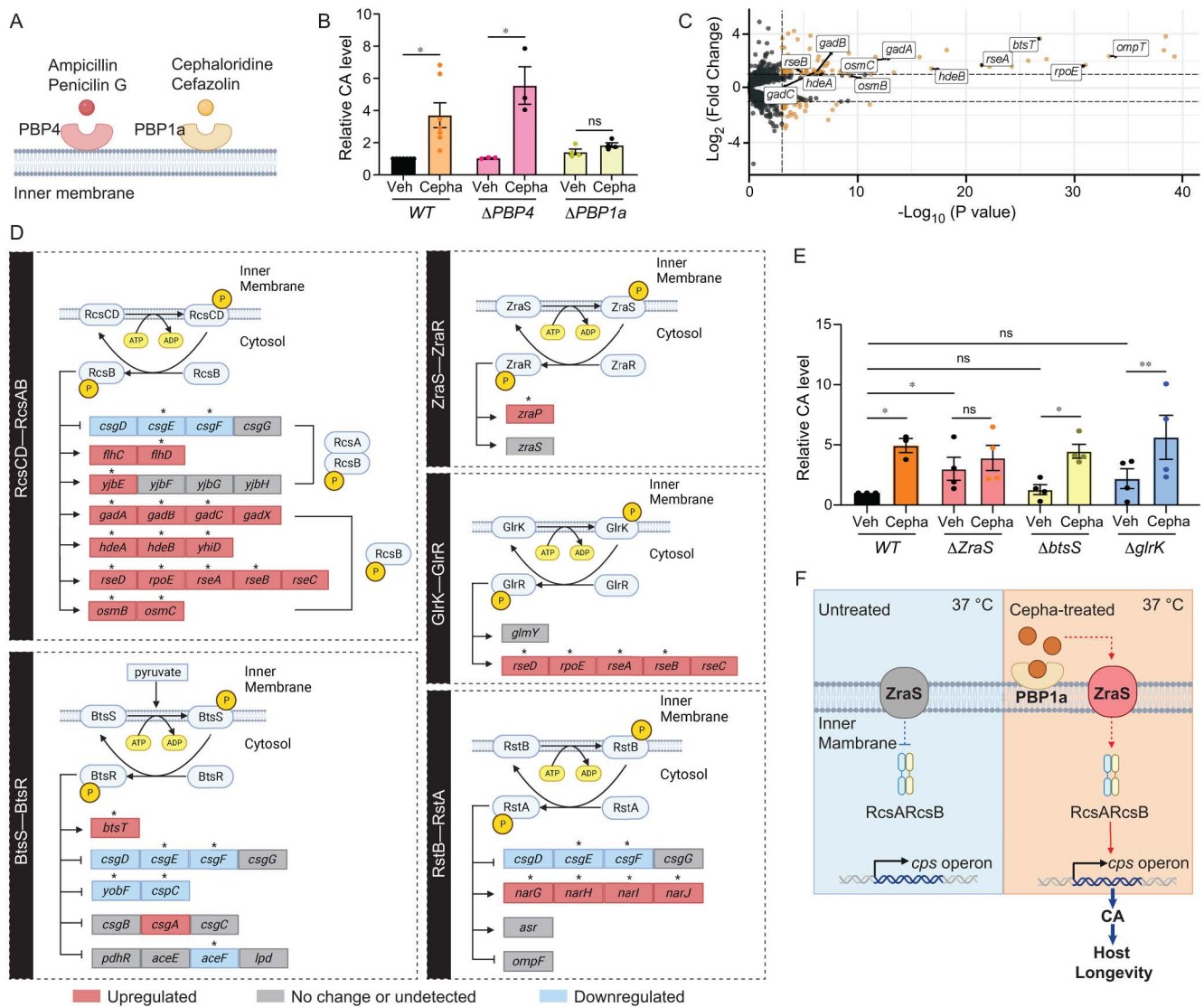

**Fig 4. PBP1a and histidine kinase ZraS mediate colanic acid (CA) chemical induction. (A)** A schematic diagram of PBP protein binding preference of different antibiotics. **(B)** The loss of PBP1a encoded by mrcA in *E. coli* suppresses CA induction upon the Cepha-CAI dose treatment. $N > 3$. **(C)** A volcano plot summarizing differentially expressed genes between the Cepha-CAI dose- and vehicle-treated *E. coli*. (Fold change > 2, FDR adjusted $p$ value < 0.05). **(D)** Known target genes regulated by the RcsB, BtsSR, ZraSR, GlrKR, and RstBA two-component systems are enriched among the genes differentially expressed in the Cepha-CAI dose-treated group. Differentially expressed genes with $\log_2$ fold change > 0.5 in red and with $\log_2$ fold change < - 0.5 in blue, * $p < 0.05$; genes with - 0.5 < $\log_2$ fold change < 0.5 or undetectable in gray. **(E)** The loss of zraS increases the CA level in the absence of cephaloridine. The Cepha-CAI dose treatment does not further increase the CA level in the zraS deletion mutant, but remains effective in either btsS or glrK deletion mutant. $N > 3$. **(F)** Illustration of the proposed model. At 37 °C, ZraS inhibits RcsAB-mediated transcription of the *cps* operon. Upon cephaloridine interacting with PBP1a, it acts through the activated ZraS to up-regulate RcsAB-mediated *cps* transcription. Ultimately, this activation promotes the production of longevity-promoting CA at 37 °C. In (B) and (E), error bars represent mean ± s.e.m., **$p < 0.01$, *$p < 0.05$, ns $p > 0.05$ by two-tailed Student $t$ test. In (C), DESeq2 |$\log_2$ fold change| ≥ 0.5; $p < 0.05$ by two-sided Wald test (cephaloridine vs. vehicle). (B, E) Source data available in S1 Data.

media. We found that the CA level in the *zraS* deletion mutant is higher than wild-type *E. coli* in the absence of the cephaloridine treatment (Fig 4E). This finding suggests that ZraS normally suppresses CA biosynthesis under baseline conditions. Furthermore, treatment with the Cepha-CAI dose did not further elevate the CA level in the *zraS* mutant (Fig 4E),

indicating that ZraS is required for mediating the CA-inducing effect of cephaloridine. In contrast, deletion of either *btsS* or *glrK* did not affect CA induction by the Cepha-CAI dose (Fig 4E). The *rstB* deletion mutant could not be evaluated in this analysis due to its impaired growth in M9 medium, even when supplemented with both carbon and nitrogen sources. Together, these results suggest ZraS functions as the inner membrane kinase that typically inhibits CA biosynthesis at 37 °C; however, exposure to low-dose cephaloridine appears to convert ZraS from an inhibitor into an activator of CA production.

In parallel, we have systematically examined the expression levels of genes from the β-lactam resistance pathway reported by KEGG pathway analysis. We found that only *ompC*, which encodes a general outer membrane porin mediating the nonspecific diffusion of small solutes, and *oppA*, which encodes the periplasmic binding protein mediating the transport of oligopeptides, are differentially up-regulated and down-regulated, respectively, in response to the Cepha-CAI dose (S3 Fig). These results suggest that the Cepha-CAI dose does not stimulate general β-lactam antibiotic stress response, which further supports that the observed induction of CA is not a secondary consequence of general antibiotic-mediated growth inhibition, but acts through a specific signaling mechanism.

Together, we propose a model (Fig 4F) that ZraS inhibits RcsAB at 37 °C, thereby silencing the *cps* operon at this temperature. Upon binding of cephaloridine to PBP1a, the activation of ZraS, which is likely associated with its autophosphorylation, triggers the transcriptional up-regulation of the *cps* operon through RcsAB and, in turn, the production of longevity-promoting CA at 37 °C.

## Discussion

In this study, we employed a xenobiotic chemical to directly target and activate a metabolic pathway of commensal *E. coli* that positively impacts the host's fitness. We showed that a low dose of cephaloridine induces CA production from commensal *E. coli*, resulting in life span extension in the host *C. elegans*. This chemical induction overcomes the temperature restriction on the *cps* gene expression in the mouse intestine. Furthermore, we discovered ZraS, a histidine kinase, as a new regulator of this chemical induction in bacteria.

Interestingly, a low dose of cephaloridine induced the *cps* operon in the mouse intestine, whereas a high dose failed to elicit the same effect. Furthermore, not all antibiotics, even other β-lactam antibiotics, exert the same effect as cephaloridine in inducing the *cps* operon. These findings indicate that the interaction between cephaloridine and bacteria is distinct from the drug's canonical antimicrobial property and differs from the drug-modifying effects of the microbiota. This highlights the significance of the chemical specificity of cephaloridine in this regulation.

Cephaloridine exhibits extremely low oral bioavailability. Its near-zero absorption in the gut has led to its discontinuation for treating infectious diseases [30]. Hence, orally supplemented low-dose cephaloridine does not directly target eukaryotic cells. It is known that the genetic product from *ampC* in *E. coli*, as well as many *Enterobacteriaceae*, has the ability to metabolize cephaloridine and other cephalosporins [37]. In wild-type *E. coli* MG1655, *ampC* expression is at a low level [38], and the low-dose cephaloridine treatment did not induce its expression in our analysis (S3 Fig). Thus, it is unlikely that cephaloridine would be metabolized by MG1655. However, we do not have direct evidence to completely exclude the possibility that cephaloridine would not be metabolized in the mouse gut by other bacteria.

Low-dose cephaloridine functions as a chemical inducer, signaling through the bacterial membrane protein PBP1a and ZraS-RcsA/B to activate the *cps* operon. Our genetic analysis suggests that ZraS suppresses CA biosynthesis at 37 °C in wild-type *E. coli*, and that low-dose cephaloridine activates ZraS, thereby relieving this suppression. In addition to its well-established role in the metal ion stress response, the ZraPSR TCS has also been linked to β-lactam antibiotic sensitivity [39]. Within the ZraPSR system, ZraP is a transcriptional target activated downstream of ZraS, and functions as a negative feedback regulator, inhibiting ZraS. A previous study reported that deletion of ZraP increases resistance to cephalothin and cefuroxime, likely through ZraR-mediated transcriptional regulation of stress response genes [39]. Although cephaloridine was not tested in that study, our findings show that cephaloridine up-regulates *zraP*, supporting

the notion that it activates ZraS signaling. Given that cephalosporin-class β-lactams, such as cephaloridine and cephalothin, primarily bind to PBP1a in *E. coli,* these findings suggest that ZraS may specifically respond to PBP1a engagement by these compounds. We propose that cephaloridine binding to PBP1a may induce a mild periplasmic perturbation, which is sensed by ZraS, leading to its autophosphorylation. As a histidine kinase, ZraS may then phosphorylate downstream RcsAB to promote *cps* operon expression and CA production. It would be interesting to investigate whether and how ZraS influences RcsAB phosphorylation in future studies. Additionally, it would be valuable to explore alternative compounds that specifically activate the ZraS-RcsAB pathway to promote CA production via the *cps* operon induction.

Our findings underscore the intricate interplay between xenobiotic chemicals, bacterial metabolism, and host fitness, urging a reevaluation of conventional drug discovery paradigms primarily focusing on eukaryotic targets. Microbiota's vast genetic diversity and metabolic pathways hold immense opportunities for developing bacteria-targeting chemical inducers or inhibitors. Further advances must rely on a systematic understanding of how microbiota-specific metabolic pathways/products influence host physiology, which will guide future drug screens focusing on microbial targets. Integrating these strategies with established microbiome-based therapeutics, as evidenced by recent studies [16], promises a synergistic approach to tackling multifaceted health challenges.

## Materials and methods

### Ethics statement

All animal procedures were performed in strict accordance with the recommendations in the *Guide for the Care and Use of Laboratory Animals* of the National Institutes of Health. For short-term Cephaloridine treatment, the protocol was approved by the Institutional Animal Care and Use Committee (IACUC) of Baylor College of Medicine, under protocol number (Protocol #AN-6975). For long-term Cephaloridine treatment, the protocol was approved by the IACUC of the Janelia Research Campus, Howard Hughes Medical Institute (Protocol #25-276). Every effort was made to minimize animal suffering and reduce the number of animals used.

### Bacteria strains

For antibiotic screening targeting *cps* transcriptional expression, we utilized the SG20781 strain (MC4100 lon+cps-B10::lacZ Mu-immλ) originally generated by S. Gottesman. The bacteria were cultured overnight at 37 °C in Luria–Broth (LB) medium and evenly plated on X-gal (40 μg/mL) supplemented LB agar plates, incubated overnight.

For CA quantification, *E. coli* Keio mutants and their parental strain BW25113 were cultured overnight at 37 °C in M9 minimum medium [40]. Bacteria were then removed by centrifugation and filtration, and the culture medium was kept for CA measurement.

For longitudinal assays, *E. coli* Keio mutants were cultivated in M9 minimum medium at 37 °C for 16 h. Subsequently, $10^7$ bacteria cells from the stationary phases were reinoculated in every 50 mL fresh M9 medium with a low dose of cephaloridine at 37 °C for another 16 h as a seeding bacterial culture. Fresh seeding bacterial cultures were made weekly. A 200 μL aliquot of the seeding bacteria culture was plated onto each 6-cm nematode growth medium (NGM) plate and maintained at 20 °C to transfer live worms every other day.

For high-resolution imaging, rph1 ilvG rfb-50 attL(Psyn135::mcherry <FRT>) attHKPR218 (cps G-R reporter strain), a genetically modified strain derived from MG1655 (genotype rph1 ilvG rfb-50), was generated by Dr. Patricia Rohs. pPR218, integrated at the HK site, encodes Pcps::sfGFP and chloramphenicol resistance.

### *C. elegans* strain

*C. elegans* var. Bristol N2 strain was obtained from the Caenorhabditis Genome Center. The strain was maintained on standard NGM agar plates seeded with corresponding bacteria at 20 °C. Age synchronization of *C. elegans* was achieved by isolating eggs as previously described [41].

## Disk-diffusion assay

*E. coli* strain SG20781 was used to assess antibiotic susceptibility conducted via disk diffusion, previously described by the Sailer group [28]. In brief, a 16-h overnight culture at 37 °C of bacteria in LB medium was spread onto LB agar containing 0.1 mg/mL 5-bromo-4-chloro-3-indolyl-β-D-galactoside (X-gal). Filter papers were saturated by 100 μL of the following antibiotic solutions (cephaloridine C258600 Toronto Research Chemicals; Cephalothin, Sigma; Cefazolin, Sigma; Cefuroxime, Sigma; penicillin, Sigma; Ampicillin, Sigma; Carbenicillin, Sigma; Colistin, Sigma; Ciprofloxacin, Sigma; Azithromycin, Sigma; Tetracycline, Sigma; Chloramphenicol, Sigma) at concentrations of 10 μg/mL and air-dried. The soaked filter papers were placed onto the bacteria-seeding plates and incubated for 24 h at 37 °C.

## Colanic acid measurement

The desired bacterial strain was first inoculated in M9 minimum medium overnight at 37 °C as the seeding culture. The OD 600 was measured, and $10^7$ cells were reinoculated in 50 mL M9 minimum medium or antibiotic-containing M9 medium for another 16 h. After the incubation, bacterial cultures were spun down at 4,000$g$ at 4 °C for 30 min. The bacterial pellet was removed, and the culture medium was filtered through a 0.2 μm syringe filter (Corning). Each 25 mL of supernatant was collected into a 50 mL conical tube, and then an equal volume of ice-cold 100% ethanol (EtOH) was used to make a final 50% EtOH for precipitation. The mixture was kept at 4 °C overnight. Once precipitation occurred, the liquid was carefully removed after centrifugation at 4,000$g$ at 4 °C for 50 min. The pellet was washed once with cold 80% EtOH and then air-dried in a hood. The pellet from every 50 mL culture was resuspended in 500 μL distilled $H_2O$. The solution was sonicated in a water bath for 60 min at 37 °C, followed by centrifugation at 4,000$g$ at 4 °C for 10 min. A 200 μL sample was boiled with 30 μL hydrochloric acid (320,331-500mL, Sigma) for 2 h. Subsequently, 60 μL 5M NaOH and 150 μL $NaHCO_3$/NaOH buffer pH 10 were added to reach pH 6–8, calibrated by 1M HCl or NaOH solutions. The fucose quantification assay was then carried out using a K-FUCOSE kit (Megazyme).

## Lifespan assay

Lifespan assays were performed as previously described [42]. In brief, age-synchronized *C. elegans* at the L1 stage were seeded onto designated bacteria lawns. Fresh bacteria-containing plates were made every other day for live *C. elegans* transfer. Once reaching adulthood as day 0, worms were transferred to new bacteria plates. Death events were scored upon each transfer. Bagged worms and vulva protruding worms were censored through the analysis. Each assay contained 80–100 animals with 30–40 animals per 6 cm plate.

## High-resolution bacteria imaging

**Imaging system.** High-resolution bacteria imaging was performed by spinning disc microscopy using a Nikon Eclipse Ti2 system and CSU-W1 SoRa confocal scanner unit under a Nikon Plan Apo λ100×/1.45 Oil lens. The excitation wavelengths were 488 nm for the GFP channel and 561 nm for the RFP channel. The images were acquired using dual camera settings in the NIS-Elements imaging platform. All the raw images were processed in the ImageJ software using customized plugin codes.

**In vitro imaging.** The *cps*G-R reporter strain was cultured in an M9 minimal medium starting from $10^7$ cells. Bacteria culture was harvested at an optical density of 600 nm of 0.6. 1 μL of the culture was directly loaded onto a 1% low-melt agarose pad made with M9 medium [43] for imaging.

**In vivo imaging.** The female C57BL/6J mice in this study were obtained from the Jackson lab and kept on a 12-h light/dark cycle with access to water and standard chow diet provided by CCM facilities. Co-housed female C57B6 mice received *cps*G-R reporter *E. coli* at 1×$10^8$ CFU/mL in 100 mL through distilled drinking water for 3 days. The bacteria-containing water was prepared and replaced daily for a continuous 3 days. On day 3, the mice received an initial 100 μL

of drug dose or sterilized water administered orally. They continued receiving drug-containing water or sterilized water for 3 h before euthanasia. Luminal content from the ileum was collected and processed based on an in vivo protocol [44] immediately after euthanasia. In brief, the ileum sections were collected and then finely minced by a pair of iris scissors in 2 mL 1× PBS, vortexed for 1 min, and then filtered with a sterile cell strainer (40 µm) to remove the tissue debris. The microbiota was washed three times with 1.5 mL PBS by centrifugation (15,000$g$, 3 min) and then resuspended in 0.1 mL PBS. 1 µL of the resuspension solution was directly loaded onto a 1% low-melt agarose pad made with M9 medium for imaging.

## SRS imaging of D-glucose incorporation

For SRS imaging [45], *E. coli* MG1655 cells were cultured overnight in M9 minimum medium at 37 °C. The overnight culture was then diluted 1:100 into 5 mL of pre-warmed M9 medium supplemented with 2% glucose-$d_7$ (Sigma-Aldrich 552003). Following incubation at 37 °C for 1, 2, 3, 4, or 6 h, 1 mL samples were collected, centrifuged, and washed twice with purified water. Approximately 6 µL of the resulting pellet was retained, resuspended, and transferred onto a microscope slide prepared with an agarose gel pad. After drying for about 2 min, a coverslip was placed on top, and the sample was ready for imaging.

SRS imaging was carried out using a custom-built SRS microscope, integrating an APE picoEmerald laser with an Olympus FV3000 confocal microscope, similar to our previously reported setup [46]. The system was carefully optimized to enhance sensitivity for the weak C-D Raman signal, enabling visualization of glucose-$d_7$ labeled bacteria. All SRS imaging was performed using a 100×/1.35 silicone oil immersion objective (Olympus UPlanSApo 100×). The pump laser was tuned to 842.2 nm and paired with a 1031.7 nm Stokes beam to excite the C-D vibration mode at 2180 cm$^{-1}$. The MG1655 cells labeled with glucose-$d_7$ were imaged at 2180 cm$^{-1}$ after various incubation times, with both control and cephaloridine-treated groups included for comparison.

## Bacterial transcriptome analysis

Bacterial cells were cultured in M9 minimal media from an initial optical density (OD$_{600}$) of 0.03 to mid-exponential phase at an OD$_{600}$ of 0.6. 1 mL of each culture was pelleted, and RNA was extracted using RNAsnap [47] and column-purified on Zymo Clean and Concentrator columns with an off-column Dnase I digestion step. Total mRNA was measured using Qubit, and the quality of RNA samples was assessed (RINe for all samples ranged from 7.9 to 8.2). Ribosomal depletion was performed with the Ribominus Transcriptome Isolation Kit. The RNA libraries were prepared for Illumina sequencing using the Illumina stranded total RNA prep kit (Catalog No. 20040525). Libraries were sequenced on an Illumina NextSeq 550 platform with 2 × 37-cycle paired-end reads. Bcl2fastq2 v2.20.0.422 was used to generate raw.fastq files, which were then filtered using FastP v0.12.4 -g –poly_g_min_len 11 -l 25. TPM calculations were done on filtered.fastq's using Salmon v1.10.0 –seqBias –gcBias –allowDoveTails to the *E. coli* MG1655 genome (NC_000913.3) with an index Kmer size of 17. DEG analysis was done on count tables using Deseq2 v1.36.0, with size factors being estimated using estimateSizeFactors() and significance assessed with nbinomWaldTest(). DEGs were visualized with pheatmap v1.0.12 and EnhancedVolcano v1.14.0 for heatmaps and volcano plots, respectively.

## Identification of *cps* operons in the gut microbiome

1. Searching *cps* operons in gut microbial genomes

The Unified Human Gastrointestinal Genome (UHGG) collection (v2.0.2) [21], comprising 289,232 nonredundant genomes from 4,744 gut prokaryotes, was downloaded from (https://www.ebi.ac.uk/metagenomics). Coding DNA sequences were extracted from the GFF files using (v0.12.8) [48] and translated into protein sequences, resulting in a redundant protein catalog containing 630,333,479 sequences. The amino acid sequences of 19 genes comprising

the cps operon [49] were used as queries to search the protein catalog using the "blastx" subcommand of DIAMOND (v2.1.10.164) [22] with an *E*-value threshold of 0.001. A total of 5,578 genomes from 47 species were found to encode all 19 *cps* genes. Protein homologs of these genes were extracted from each genome and aligned using MAFFT (v7.526) [50]. The resulting 19 multiple sequence alignments (MSAs) were trimmed with trimAl (v1.5. rev0) [51] and concatenated using the "concat" subcommand of SeqKit (v2.9.0) [52]. The concatenated MSA was used to construct a phylogenetic tree in IQ-TREE (v2.3.6) [53] with the parameters "-bb 1000 -alrt 1000 –asr." The tree was visualized using the ggtreeExtra R package (v1.6.1) [54].

2. Searching *cps* operons in gut metagenomes

Two publicly available metagenomic datasets of centenarians were obtained from the European Nucleotide Archive (ENA) under accession numbers PRJNA675598 and PRJEB25514. The first dataset, from a Japanese cohort ($N = 265$), included 119 centenarians, 107 elderly, and 39 young individuals [23]. The second dataset, from an Italian cohort ($N = 59$), included 19 centenarians, 23 elderly, and 17 young individuals [24]. Metagenomic reads were aligned to the *cps* operon using BWA (v0.7.18) [arXiv:1303.3997v2]. SAM files were converted to BAM format, and the number of mapped reads was quantified using Samtools (v1.21) [55].

## Metagenomic sequencing

Microbial genomic DNA was extracted from 48 mice fecal pellets using the ZymoBiomics DNm Mini Prep kit (p/n D4300), with bead beating performed using the tube adapter from Zymo for the Vortex Genie 2 (p/n S5001-7), along with a 75 µL aliquot of the ZymoBIOMICS Microbial Community Standard (p/n D6300) as a positive control, and a 75 µL aliquot of Zymo Shield (p/n R1100) as a negative control. DNA were quantified using a Qubit 3.0 fluorometer (p/n Q33216), broad-range DNA assay (p/n Q32853) from Thermo Fisher. Samples were prepared for sequencing using 1 ng gDNA inputs into the Illumina Nextera XT library preparation kit following the manufacturer's recommendations. Resulting libraries were quantified using the Qubit 3.0 fluorometer high-sensitivity DNA assay (p/n Q32854) and size-qualified using the Agilent Bioanalyzer high-sensitivity DNA assay (p/n 5067−4626). Libraries were equimolarly pooled and then sequenced using a NextSeq 2000 P3-300 XLEAP kit (p/n 20100988).

## Metagenomic data analyis

1. Data processing and taxonomic profiling

Raw sequencing data in FASTQ format were processed using Taxprofiler (v1.0.0), a reproducible workflow for taxonomic profiling of metagenomic samples (https://www.biorxiv.org/content/10.1101/2023.10.20.563221v1; https://github.com/nf-core/taxprofiler). This pipeline facilitates standardized and efficient processing of metagenomic data through an automated workflow that integrates quality control, preprocessing, and multiple taxonomic classification tools.

For taxonomic classification, we employed MetaPhlAn4 [56], which uses clade-specific marker genes to identify and quantify microbial species present in the samples. This approach enables accurate resolution of bacterial, archaeal, viral, and eukaryotic microbes at species-level taxonomic resolution. The resulting taxonomic profiles were organized into abundance tables for downstream analysis.

2. Microbiome diversity analysis

The taxonomic abundance data were analyzed using the phyloseq package [57] in R. Alpha diversity metrics were calculated using the vegan package(https://vegandevs.github.io/vegan/), which is integrated with phyloseq for comprehensive microbiome analysis.

From the taxonomic profiles, we calculated three alpha diversity metrics to assess the diversity within each sample:

*Shannon Diversity Index*: Calculated using the vegan::diversity function, this composite measure accounts for both species richness and evenness, with higher values indicating greater diversity.

*Pielou's Evenness*: Derived by dividing the Shannon diversity index by the natural logarithm of species richness (Shannon/log(S)), this metric quantifies how evenly individuals are distributed among different species, with values ranging from 0 (complete dominance by a single taxon) to 1 (perfect evenness).

*Richness*: Determined using vegan::estimateR, this represents the observed number of unique taxa in each sample.

3. Statistical analysis

To evaluate the effects of experimental conditions on microbial diversity during analysis, mice were assigned to either Control or Experimental groups, with measurements taken at both pretreatment and posttreatment timepoints. This created four distinct groups for comparison: Control-Pre treatment, Control-Post treatment, Experimental-Pre treatment, and Experimental-Post treatment. We performed Tukey HSD (Honest Significant Difference) tests following analysis of variance (ANOVA) to evaluate differences across all four treatment groups, with appropriate corrections for multiple comparisons.

To assess differences in *cps* operon abundance between centenarians and elderly controls, Welch's *t* tests were used to compare $log_{10}$-transformed cps abundance between centenarians and elderly controls, following confirmation of normality (Shapiro–Wilk test $p > 0.05$). Effect sizes were quantified using Cohen's *d*, and statistical power was evaluated by estimating the required sample size to detect the observed effect size with 80% power at a significance level of 0.05.

**Long-term cephaloridine studies in mice**

1. Treatment procedure

C57BL/6J mice (Strain #000664) were obtained from The Jackson Laboratory and maintained on PicoLab Rodent Diet 20 (LabDiet, 5053). Cephaloridine (2 µg/mL cephaloridine C258600 Toronto Research Chemicals) and *E. coli* ($1 \times 10^8$ CFU/mL, MG1655) were administered via drinking water. The control group followed a weekly schedule consisting of 24 h on *E. coli*-supplemented water, 54 hours on regular water, another 24 h on *E. coli*-supplemented water, and 66 h on regular water. In contrast, the cephaloridine treatment group followed an identical schedule, with cephaloridine-supplemented water substituted for regular water during the 54- and 66-h intervals.

2. Fecal collection

Fresh fecal samples were collected from individual mice, placed in pre-labeled sterile tubes. All fecal samples were immediately snap-frozen in liquid nitrogen and stored at −80 °C until processing for microbial DNA extraction and further shotgun-metagenomic sequencing.

3. Blood collection

Blood was collected via submandibular vein bleeding using lithium heparin tubes (BD 365965). Samples were centrifuged at 2,000*g* for 15 min at 4 °C to separate plasma, which was stored at −0 °C for subsequent metabolic analysis.

4. Plasma metabolic panel analysis

All the assays were performed according to the manufacturer's protocol using either ELISA-based or end point colorimetric assay system in a 96-well microplate format. Standard curves were generated for individual assays using the provided standards to determine the concentration of unknown samples. Plasma levels of adiponectin, leptin, and insulin were quantified using ELISA kit (Millipore). Non-esterified free fatty acids (NEFAs) were determined using Wako NEFA colorimetric assay system (Fujifilm). Glucose, triglycerides, and total cholesterol were detected using colorimetric assay kits

 

PLOS Biology

(Abcam). Plasma HDL and LDL levels were detected using ELISA kit (LS Bio). Glycerol concentrations were measured using an endpoint colorimetric assay (Sigma-Aldrich). Absorbance readings were acquired using Benchmark Plus microplate reader (Bio-Rad). Concentrations of the samples were automatically calculated based on the corresponding standard curve used for the assays.

## Statistical analysis

For all figure legends, asterisks indicate statistical significance as follows: NS, not significant ($p > 0.05$), *$p < 0.05$, **$p < 0.01$, ***$p < 0.001$, and ****$p < 0.0001$. Data were obtained by performing independently at least three biological replicates unless specified in the figure legends. No statistical method was used to pre-determine the sample size. No data were excluded from the analyses. Two-tailed Student $t$ test or one-way or two-way ANOVA with Holm–Sidak corrections were used as indicated in the corresponding figure legends. $N$ indicates the number of biological replicates. $N$ indicates the number of animals or technical replicates within each biological replicate. For survival analysis, statistical analyses were performed with SPSS software (IBM) using Kaplan–Meier survival analysis and log-rank test, as well as using GraphPad Prism 10 survival analysis. For RNA-seq, a two-sided Wald test in R package Deseq2 was used. Figures and graphs were constructed using BioRender.com, ImageJ, GraphPad Prism 10 (GraphPad Software), and Illustrator (CC 2019; Adobe).

## Supporting information

**S1 Fig.** Related to Fig 1. **(A)** A summary of *cps* induction indicated by the intensity and the size of the blue coloration in WT BW25113 *E. coli* treated with different antibiotics. **(B)** Measurement of carbon-deuterium levels in WT *E. coli* cultured in M9 medium supplemented with deuterium-labeled glucose using stimulated Raman scattering (SRS) microscopy at 2180 cm$^{-1}$, indicating no metabolic suppression by the Cepha-CAI dose. 30–60 bacterial cells per condition. Error bars represent the mean ± s.d., *ns p > 0.05* by Two-way ANOVA. Source data available in S1 Data.
(EPS)

**S2 Fig.** Related to Fig 2. **(A)** Representative ratiometric images of *E. coli* from the mouse gut, vehicle verse 2 µg/mL cephaloridine (left), vehicle verse 4 and 8 µg/mL cephaloridine verse tetracycline 10 µg/mL (right), scale bar = 5 µm. **(B)** Schematic diagram presenting the long-term drug administration procedure in wild-type B6 mice. **(C–K)** Plasma levels of low-density lipoprotein (LDL) (C), high-density lipoprotein (HDL) (D), triglyceride (E), leptin (F), adiponectin (G), insulin (H), glucose (I), free fatty acids (FFA) (J), and glycerol (K) were compared between 12 and 18 months of age in both male and female mice treated orally with water (vehicle) or cephaloridine (2 µg/mL). Source data available in S1 Data. **(L)** Pielou's evenness of the fecal microbiota in 12- and 18-month-old mice. Lines join paired samples from the same mouse taken before (12 m) and after (18 m) a 6-month course of vehicle (left) or Cephaloridine (right) treatment. One-way ANOVA across the Cephaloridine before and after treatment groups revealed no significant differences. Source data available in S1 Data. **(M)** Number of observed amplicon-sequence variants (ASVs) in the same mice and time-points as in (L), shown as a measure of microbial richness. Statistical analysis identical to (L); no significant differences were detected after Cephaloridine treatment. Source data available in S1 Data. **(N)** Shannon diversity index, integrating richness and evenness, for the fecal microbiota of the four treatment groups described above. One-way ANOVA with Tukey's HSD indicated no significant changes before and after Cephaloridine treatment. Source data available in S1 Data.
(EPS)

**S3 Fig.** Related to Fig 4. An illustration of the β-lactam antibiotic resistance pathway in MG1655 *E. coli* adopted from KEGG (https://www.genome.jp/pathway/ko01501), which is color-coded based on the RNA-seq analysis of gene

expression between Cepha-CAI dose-treated and vehicle-treated *E. coli*. Differentially expressed genes with $\log_2$ fold change $> 0.5$ in red and with $\log_2$ fold change $< -0.5$ in blue, * $p < 0.05$; genes with $-0.5 < \log_2$ fold change $< 0.5$ or undetectable in gray.
(EPS)

**S1 Table.** Summary of life span analyses shown in Figs 1J, 2B, 3C, and 3D.
(XLSX)

**S1 Data.** Source data. Summary of all raw data underlying the quantification analyses presented in main and supplementary figures. Figs 1B, 1C, 1E–1H, 1J, S1B, 2A, 2B, 2E, 2G–2J, S2C–S2N, 3B–3D, 4B, and 4E.
(XLSX)

**S2 Data.** Tree files for the phylogenetic trees in Fig 1A.
(TREE)

**S3 Data.** Codes for conducting analyses in Fig 2G.
(PY)

**S4 Data.** Codes for conducting analyses in Fig 1A–1C.
(RMD)

## Acknowledgments

We thank Dr. J. Petrosino, Dr. T. Palzkill, Dr. W. Craigen, and Dr. D. Moore for invaluable guidance and discussions on this project. We express strong gratitude to Dr. P. Rohs for generating the *cps*G-R reporter strain and Dr. T. Chen for generating the analytical codes for bacteria fluorescence intensity analysis. We think the expert assistance of Dr. C. Ward and Dr. P. Saha at the Mouse Metabolism and Phenotyping Core at Baylor College of Medicine, which was funded by NIH (UM1HG006348, R01DK114356, R01HL130249). We thank Quantitative Genomics Support Team at Janelia and the expert assistance of Dr. P. Chung and Dr. C. Raley.

## Author contributions

**Conceptualization:** Guo Hu, Marzia Savini, Matthew Brandon Cooke, Xin Wei, Dinghuan Deng, Alice X. Wen, Jin Wang, Chao Jiang, Christophe Herman, Jiefu Li, Meng C. Wang.

**Data curation:** Guo Hu, Marzia Savini, Matthew Brandon Cooke, Xin Wei, Dinghuan Deng, Youchen Guan, Alice X. Wen, Xin Yu, Jin Wang.

**Formal analysis:** Guo Hu, Marzia Savini, Xin Wei, Dinghuan Deng, Shihong M. Gao, Ruyue Alps Xia.

**Funding acquisition:** Christophe Herman, Meng C. Wang.

**Investigation:** Guo Hu, Marzia Savini, Xin Wei, Dinghuan Deng, Youchen Guan.

**Methodology:** Guo Hu, Xin Wei, Dinghuan Deng, Shihong M. Gao, Jin Wang, Chao Jiang, Meng C. Wang.

**Project administration:** Guo Hu.

**Resources:** Guo Hu, Jin Wang, Christophe Herman, Meng C. Wang.

**Software:** Guo Hu, Xin Wei, Ruyue Alps Xia.

**Supervision:** Chao Jiang, Christophe Herman, Jiefu Li, Meng C. Wang.

**Validation:** Guo Hu, Meng C. Wang.

**Visualization:** Guo Hu, Marzia Savini, Xin Wei, Dinghuan Deng, Shihong M. Gao, Ruyue Alps Xia, Meng C. Wang.

**Writing – original draft:** Guo Hu, Marzia Savini, Xin Wei.

**Writing – review & editing:** Guo Hu, Marzia Savini, Matthew Brandon Cooke, Xin Wei, Ruyue Alps Xia, Alice X. Wen, Jin Wang, Christophe Herman, Meng C. Wang.

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
