## [Editor Report · Decision Letter 0]

10 Jul 2024

Dear Meng,

Thank you for submitting your manuscript entitled "Chemical Induction of Longevity-Promoting Colanic Acid in the Host’s Microbiota" for consideration as a Short Report by PLOS Biology.

Your manuscript has now been evaluated by the PLOS Biology editorial staff as well as by an academic editor with relevant expertise and I am writing to let you know that we would like to send your submission out for external peer review.

Once your full submission is complete, your paper will undergo a series of checks in preparation for peer review. After your manuscript has passed the checks it will be sent out for review. To provide the metadata for your submission, please Login to Editorial Manager (https://www.editorialmanager.com/pbiology) within two working days, i.e. by Jul 12 2024 11:59PM.

Kind regards,

Ines

--

Ines Alvarez-Garcia, PhD

Senior Editor

PLOS Biology

---

## [Decision Letter · Decision Letter 1]

17 Sep 2024

Dear Dr Wang,

Thank you for your patience while your manuscript entitled "Chemical Induction of Longevity-Promoting Colanic Acid in the Host’s Microbiota" was peer-reviewed at PLOS Biology as a Short Report. Your manuscript has been evaluated by the PLOS Biology editors, an Academic Editor with relevant expertise, and by three independent reviewers.

The reviews are attached below. As you will see, the reviewers find the conclusions novel and interesting, but they also raise several concerns that would need to be addressed before we can consider the manuscript for publication. Reviewer 1 is not convinced with the evidence suggesting a role of colonic acid in mice and thinks that the description of the methods should be improved and that you would need to clarify the role in the context of host longevity and health. In that line, Reviewer 2 asks for clarification of the physiological benefits of the CA induction in the mouse experiments and makes several suggestions to improve other experiments. Reviewer 3 also thinks that the data on the lifespan extension induced by antibiotic treatment should be clarified and strengthened, and that better evidence should be provided to confirm that the dosages used can act as anti-microbial in vivo.

Based on their specific comments and following discussion with the Academic Editor, it is clear that a substantial amount of work would be required to meet the criteria for publication in PLOS Biology. However, given our and the reviewer interest in your study, we would be open to inviting a comprehensive revision of the study that thoroughly addresses all the reviewers' comments. Given the extent of revision that would be needed, we cannot make a decision about publication until we have seen the revised manuscript and your response to the reviewers' comments. Your revised manuscript would need to be seen by the reviewers again, but please note that we would not engage them unless their main concerns have been addressed.

We appreciate that these requests represent a great deal of extra work, and we are willing to relax our standard revision time to allow you 6 months to revise your study. Please email us (plosbiology@plos.org) if you have any questions or concerns, or envision needing a (short) extension.

**IMPORTANT - SUBMITTING YOUR REVISION**

3. Resubmission Checklist

a) *PLOS Data Policy*

b) *Published Peer Review*

Sincerely,

Ines

--

Ines Alvarez-Garcia, PhD

Senior Editor

PLOS Biology

Reviewers' comments

Rev. 1:

This manuscript investigates a novel approach to modulating gut microbiota to enhance host longevity through chemical induction of bacterial metabolites. Specifically, the study explores how low doses of the antibiotic cephaloridine can induce the production of colanic acid (CA) in Escherichia coli (E. coli), which has been previously shown to extend the lifespan of the model organism Caenorhabditis elegans (C. elegans). The authors demonstrate that cephaloridine, at sub-minimum inhibitory concentrations, triggers the overproduction of CA by activating the cps operon in E. coli, even at temperatures above 30°C, which typically inhibit CA production. They extend their findings to the mammalian system by showing that low-dose cephaloridine can induce CA production in the mouse gut, suggesting potential translational relevance. The study also uncovers a novel mechanism for CA induction, involving the membrane-bound histidine kinase ZraS, independent of the RcsC-RcsD two-component system traditionally associated with CA biosynthesis.

While the study is well-conducted and presents intriguing findings, the manuscript's message could be misleading or incomplete in certain aspects.

Unclear Role of CA in Mice:

The manuscript does not adequately address the role of CA in mice, which is a significant gap considering that the study aims to explore the translational potential of CA induction from a microbial metabolite to a mammalian system. The authors need to clarify what is known about CA's effects in mammals, especially in the context of host longevity and health.

State-of-the-Art in Microbial Metabolic Pathway Activation:

The introduction lacks sufficient detail on the current understanding of drug-induced activation of microbial metabolic pathways in both commensals and pathogens. This background is crucial for contextualizing the novelty and significance of the study's approach.

The manuscript's intention to suggest a potential beneficial effect of CA in the host might be premature, given the limited evidence presented for its role in mammals. The most compelling part of the study is the discovery of a non-canonical mechanism for CA induction via cephaloridine. However, if the focus shifts solely to this mechanistic insight, the manuscript might not align with the broader biological focus typically expected by a journal like PLoS Biology.

Overall, the study presents a novel and interesting method for chemically inducing bacterial metabolites that may have beneficial effects on host organisms. However, the manuscript would benefit from a more transparent discussion of the role of CA in mammals and a stronger contextualization of the study within the broader field of microbiome-based therapeutics. The authors should consider focusing on the mechanistic novelty of the CA induction pathway, although this may require reassessing the manuscript's suitability for PLoS Biology.

Rev. 2:

This is a review of the manuscript "Chemical Induction of Longevity-Promoting Colanic Acid in the Host's Microbiota" by Hu and colleagues for PLoS Biology. It previously has been shown that the polysaccharide Colanic Acid (CA) extends the lifespan of C. elegans and Drosophila. CA is produced by E. coli and related microbiota that can be found in the gut. In previous work, the authors used optogenetics to increase production of CA by gut microbiota, demonstrating that this was sufficient to induce longevity.

Here now the authors describe a chemical approach to induce CA production in the gut, by exposing animals to a low dose of cephaloridine. Cephaloridine is an antibiotic, but the dose used by the authors is below of what would be required to impair the bacterial growth. The authors show that this approach induces CA production in the gut of C. elegans and mice. Further, they show in C. elegans that such treatment is sufficient to induce longevity. Finally, they delineate the mechanism by which cephaloridine induces CA production in E. coli. They argue based on genetic experiments that chephaloridine acts on the bacterial membrane protein PBP1a which in turn relieves an inhibitory effect of the kinase ZraS on the RcsA/RcsB system. Then active RcsA/RcsB will activate transcription of the cps operon which encodes enzymes needed for CA production.

Overall, this is a solid story for a short report. The experiments have been well conducted and eventually reveal a new concept for how longevity could be promoted in vivo, using a drug that does not act on the organism of interest itself but that rather modulates the metabolism of its gut microbiota.

However, the following specific points should still be addressed:

1) You mention that CA is produced by E. coli and also other Enterobacteriaceae. Do you assume that the same mechanism of cepha-induced CA production exists in these other species, too? If so, what fraction of the human gut microbiota would actually be E. coli and what fraction those other Enterobacteriaceae? It would be valuable to clarify these points in the manuscript.

2) Line 162: You found before that a dose of 1.8 µM is optimal. Why did you then use here only higher doses of 4 and 8 µM? This is not clear to me.

3) You primarily use growth curves to argue that the low Cepha doses don't give antibiotic effects. Given that in its function as an antibiotic Cepha targets the cell wall, it would be good to see directly that the cell remains unaffected by these low doses. Can you somehow directly monitor the condition and integrity of the cell wall?

4) Line 260: Your referral to Figure 4J does not make sense for me. Please double check.

5) Line 260: It is not clear to me why you consider ZraS to be responsible for the inhibition of CA production above 30°C and that Cepha releases this inhibitory effect. What is the evidence for that? This is a crucial part of the mechanism that should be addressed or clarified.

6) Likewise, how would binding of Cepha to PBP1a act on ZraS? Please give some better insight into this.

7) What are the physiological benefits of the CA induction by Cepha in your mouse experiments? Could you see any? Please comment in the manuscript. If you saw any, they would be a great addition to this manuscript.

8) Line 301: What is the evidence that bacteria don't metabolize the Cepha? Is there a reference showing this?

Rev. 3:

Hu te al. present an intriguing investigation into the induction of colanic acid (CA) by E. coli both in culture and in vivo in the mouse gut. Previously, the group had identified CA as a lifespan-extending compound as part of a larger screen for microbial modulators of C. elegans aging. Here, they aim to use E. coli to deliver Colanic acid to the guts of mice.

The work is careful and well-considered—the authors discover new aspects of E. coli gene regulation and metabolism while finding some success in their overall synthetic biology / engineering goal. The methods they use are compelling, a mixture of E. coli bio-engineering and in vivo experimentation in nematodes and mice. However, the authors do not appear to provide direct evidence that they have achieved "chemical induction of longevity-promoting Colanic acid in the host's microbiota" in either C. elegans or mice. In C. elegans, the authors rely on lifespan as a proxy for CA production, but this proxy needs more validation (see point 1). In mice, the authors rely on a cleverly-engineered bacterial reporter for cps promoter activity which again needs additional validation. This reviewer does not think that direct measurement of CA induction in the gut is strictly required for the authors' results to be convincing—though such measurements would be helpful, definitive proof.

Major points

1. The authors show that treatment with CA at the CAI dosage extends C. elegans lifespan by 14%. This is a relatively small effect for nematode lifespan experiments—smaller than the reciprocal lifespan shortening the authors previously saw produced by Δlon and ΔHns mutants. What is the evidence that the lifespan extension produced by antibiotic treatment is specifically mediated by CA production? Could the cephaloridine be acting directly on some target in C. elegans to produce this small lifespan extension independently of its influence on E. coli?

2. The authors do not see an increase in RFP induction at the higher 8 ug/mL antibiotic dosage. The authors' favoured model seems to be that cephaloridine at this dose acts as an anti-microbial in vivo, unlike at 4 ug/mL. However, the authors provide no evidence for this. To clarify the missing effect at 8 ug/uL, the authors should provide some evidence that 4 ug/mL and 8 ug/mL dosages fall into distinct regimes in respect to anti-microbial activities. Otherwise, the reader is left wondering whether the experiments suffer from some technical issue—for example that the difference between 4 ug and 8 ug per mL results from inconsistent batch-to-batch variation and not a true dose-dependent qualitative difference in cephaloridine activity.

Minor points

1. In describing previous work, the authors state that "[CA] cannot be effectively delivered through the GI tract". Absent additional explanation, this fact seems to preclude the author's stated goal of using E. coli to deliver CA via the GI tract. The authors should give some more context or explanation. A citation would also help readers to understand the background for these statements.

2. The authors state that the most effective CA-inducing concentration of cephaloridine is 1.8 ug/mL, which they claim is lower than a previously reported MIC value of 2 ug/mL. 1.8 does not seem to be meaningfully lower than 2. Do the authors have some statistical analysis to suggest the 0.2 ug/mL difference can be interpreted in this way? An straightforward solution would be for the authors to directly compare the CAI and MIC dosages in a single experiment under controlled conditions.

3. The authors state that rtsB deletion inhibits bacterial growth in "M9 culture medium". The conventional recipe for this medium lacks both carbon and nitrogen sources. So growth wouldn't be expected by any E. coli strain, no?

4. The authors state that their CAI dosage is "unlikely to stimulate the general beta-lactan antibiotic stress response". The authors provide evidence for this, but the reader is left wondering why this observation is important. Can the authors explain a bit more?

5. The authors identify an 8% increase in GFP-to-RFP ratios from their in vivo experiments (Fig 2G), a smaller effect than the 20% they see in liquid culture (Fig 2E). Can the authors explain this difference?

6. It is great to see the authors learning something new about E. coli metabolism in the context of this larger engineering effort. How does the observed regulation of CA by ZraS mesh with the other known physiologic activities of ZraS? Some more perspective would be interesting if space allows.

---

## [Decision Letter · Decision Letter 2]

14 Jul 2025

Dear Dr Wang,

Thank you for your patience while we considered your revised manuscript entitled "Chemical Induction of Longevity-Promoting Colanic Acid in the Host’s Microbiota" for consideration as a Short Report at PLOS Biology. Your revised study has now been evaluated by the PLOS Biology editors, the Academic Editor and the three original reviewers.

The reviews are attached below. In light of the comments, we are pleased to offer you the opportunity to address the remaining points from Reviewers 2 and 3 in a revision that we anticipate should not take you very long. We will then assess your revised manuscript and your response to the reviewers' comments with our Academic Editor aiming to avoid further rounds of peer-review, although we might need to consult with the reviewers, depending on the nature of the revisions.

**IMPORTANT - SUBMITTING YOUR REVISION**

3. Resubmission Checklist

a) *PLOS Data Policy*

b) *Published Peer Review*

Sincerely,

Ines

--

Ines Alvarez-Garcia, PhD

Senior Editor

PLOS Biology

Reviewers' comments

Rev. 1:

Accept.

Rev. 2:

I am satisfied by the revisions made by the authors, with one minor exception:

Regarding my point 1), you responded: "We appreciate the reviewer's question and suggestion. Enterobacteriaceae are present in healthy human commensals with a small fraction [PMID: 26499895]. Within the Enterobacteriaceae family, Escherichia is the most dominant taxa in healthy humans [PMID: 20157339]. However, the presence of cps operon, which is responsible for CA production, in human commensal has not been previously analyzed. We have examined the presence of cps operon in the Unified Human Gastrointestinal Genome (UHGG) database. We detected the existence of the cps operon in over 5000 genomes across 47 specifies, such as Escherichia (including E. coli), Salmonella, Enterobacter, and Citrobacter (new Figure 1A). Furthermore, we compared the presence of the cps operon between centenarians and their elderly controls using two publicly available human centenarian gut microbiome datasets (PRJNA675598, PRJEB25514). We found that the read count of the cps operon was significantly higher in centenarians compared to controls in the Japanese gut microbiome dataset (PRJNA675598), which includes more than 100 individuals in each group (p=0.015, new Fig. 1B). A similar trend was also observed using the Italian gut microbiome dataset (PRJEB25514; p=0.087), although it did not reach statistical significance, likely due to the smaller sample size, with only ~20 individuals in each group (new Fig. 1C). These analyses revealed a potential correlation between the CA overproduction and human longevity. We have included these new analyses in the revised manuscript."

My follow-up: It's great to hear that the cps operon exists also beyond the Enterobacteriaceae family. However, can you please provide the readers with actual or at least approximate numbers on what fraction of human commensals tends to be Enterobacteriaceae and what fraction tend to be bacteria carrying the cps operon? Just saying "small franction" or "5000 genomes across 47 species" does not give me a true sense. I think that the readers would appreciate this information.

Rev. 3:

The authors have addressed all of my original concerns, which is very good.

I have only one major concern with the revised manuscript: I find the new human microbiome sequence data to be entirely unconvincing. The difference between centenarians and control in the Italian cohort is not statistically significant, which many would interpret as supporting the opposite conclusion reached by the authors: that centenarians do not differ in the gut microbiome composition at this locus. The authors deviate from statistical best practices by speculating their negative result is "likely due to the smaller sample size, with only ~20 individuals in each group". This is not correct. The data demonstrates clearly that no large differences exist between the subgroups in the Italian cohort. Only differences small enough to be invisible when measured using 20 individuals might still exist despite the data presented. The best the authors can do here is perform a power calculation that estimates the minimum effect size identifiable from these 20 individuals—this minimum then sets the ceiling on the maximum difference that might exist. Would this small magnitude then retain the potential to be convincing? The authors have much more solid data elsewhere in their manuscript--it's unclear why the authors attempt to add further support to their claims using additional data lacking statistically significant effects.

Even in the Japanese cohort, where the difference between centenarians and control does reach a permissive definition of statistical significance, the effect size is not stated but is clearly very small. For their data to be convincing, the authors would need to explain how this small statistical difference between groups reflects biologically meaningful difference, for example. Why should a small difference at this locus be expected to have any effect at all? Furthermore, an effect size this small would seem to be possibly generated by any of the various potential confounders that likely to exist between these two groups--do the authors try to control for diet, lifestyle, and age as potential confounders that might directly modulate cps frequencies? Doesn't it seem more likely that the measured difference in cps opron abundance result as the downstream effect of the many differences in lifestyle, diet, medication consumption, and aged gut microenvironment of centenarians compared to control populations? The likely causal model would then be (being old) -> (cps operon abundance), and not the authors favored model (cps operon abundance) -> (reaching old age).

My recommendation to the authors would be to remove all the human data, as it detracts from an otherwise convincing model organism project. Alternatively, the authors should adhere to statistical best practices and conclude that they observe a very small difference (stating the effect size) in one out of two cohorts of centenarians, while addressing the potential for any such differences to result from a variety of potential confounds.

---

## [Editor Report · Decision Letter 3]

20 Aug 2025

Dear Dr Wang,

Thank you for your patience while we considered your revised manuscript "Chemical Induction of Longevity-Promoting Colanic Acid in the Host’s Microbiota" for publication as a Short Reports at PLOS Biology. This revised version of your manuscript has been evaluated by the PLOS Biology editors, and the Academic Editor.

Based on our Academic Editor's assessment of your revision, we are likely to accept this manuscript for publication, provided you satisfactorily address the following data and other policy-related requests:

a) We routinely suggest changes to titles to ensure maximum accessibility for a broad, non-specialist readership, and to ensure they reflect the contents of the paper. In this case, we would suggest a minor edit to the title, as follows. Please ensure you change both the manuscript file and the online submission system, as they need to match for final acceptance:

"Chemical modulation of gut bacterial metabolism induces colanic acid and extends the lifespan of nematode and mammalian hosts"

b) As your work has been done using mice, we do require an ethics statement. The Ethics statement needs to be a separate, independent (and the first) subheading in the Material & Methods section. It must include the full name of the IACUC/ethics committee that reviewed and approved the animal care and use, as well as the protocol/permit/project license number. https://journals.plos.org/plosbiology/s/ethical-publishing-practice

c) Thank you for already providing the source data for most figure. However we still require the values for figures: 1BJ, 2B, 3CD, S2LMN

d) Please cite the location of the data clearly in all relevant main and supplementary Figure legends, e.g. “The data underlying this Figure can be found in S1 Data” or “The data underlying this Figure can be found in https://doi.org/10.5281/zenodo.XXXXX”

e) Supplementary files (e.g., excel). Please ensure that all data files are uploaded as 'Supporting Information' and are invariably referred to (in the manuscript, figure legends, and the Description field when uploading your files) using the following format verbatim: S1 Data, S2 Data, etc. Multiple panels of a single or even several figures can be included as multiple sheets in one excel file that is saved using exactly the following convention: S1_Data.xlsx (using an underscore).

f) Please provide the tree files for the phylogenetic trees in Figures 1A. Please make sure all relevant figures have scale bars.

g) Please ensure that your Data Statement in the submission system accurately describes where your data can be found and is in final format, as it will be published as written there

h) Thank you for providing the underlying code in GitHub. However, please note that we cannot accept sole deposition of code in GitHub, as this could be changed after publication. However, you can archive this version of your publicly available GitHub code to Zenodo. Once you do this, it will generate a DOI number, which you will need to provide in the Data Accessibility Statement (you are welcome to also provide the GitHub access information). See the process for doing this here: https://docs.github.com/en/repositories/archiving-a-github-repository/referencing-and-citing-content

We expect to receive your revised manuscript within two weeks.

*Published Peer Review History*

*Press*

Sincerely,

Melissa

Melissa Vázquez Hernández, PhD

Associate Editor

PLOS Biology

on behalf of

Ines

Ines Alvarez-Garcia, PhD

Senior Editor

PLOS Biology

---

## [Editor Report · Decision Letter 4]

19 Sep 2025

Dear Dr Wang,

Thank you for the submission of your revised Short Report entitled "Chemical Modulation of Gut Bacterial Metabolism Induces Colanic Acid and Extends the Lifespan of Nematode and Mammalian Hosts" for publication in PLOS Biology. On behalf of my colleagues and the Academic Editor, Mark Alkema, I am delighted to let you know that we can in principle accept your manuscript for publication, provided you address any remaining formatting and reporting issues. These will be detailed in an email you should receive within 2-3 business days from our colleagues in the journal operations team; no action is required from you until then. Please note that we will not be able to formally accept your manuscript and schedule it for publication until you have completed any requested changes.

PRESS

Sincerely, 

Ines

--

Ines Alvarez-Garcia, PhD

Senior Editor

PLOS Biology
